# Palmitoylation of ULK1 by ZDHHC13 plays a crucial role in autophagy

Keisuke Tabata [1,2], Kenta Imai[1,2], Koki Fukuda[1,2], Kentaro Yamamoto[1,2], Hayato Kunugi[1,2], Toshiharu Fujita[1,2], Tatsuya Kaminishi [2,3], Christian Tischer[4], Beate Neumann [4], Sabine Reither[4], Fatima Verissimo[4], Rainer Pepperkok[4,5], Tamotsu Yoshimori [1,2,3] ✉ & Maho Hamasaki [1,2] ✉

Autophagy is a highly conserved process from yeast to mammals in which intracellular materials are engulfed by a double-membrane organelle called autophagosome and degrading materials by fusing with the lysosome. The process of autophagy is regulated by sequential recruitment and function of autophagy-related (Atg) proteins. Genetic hierarchical analyses show that the ULK1 complex comprised of ULK1-FIP200-ATG13-ATG101 translocating from the cytosol to autophagosome formation sites as a most upstream ATG factor; this translocation is critical in autophagy initiation. However, how this translocation occurs remains unclear. Here, we show that ULK1 is palmitoylated by palmitoyltransferase ZDHHC13 and translocated to the autophagosome formation site upon autophagy induction. We find that the ULK1 palmitoylation is required for autophagy initiation. Moreover, the ULK1 palmitoylated enhances the phosphorylation of ATG14L, which is required for activating PI3-Kinase and producing phosphatidylinositol 3-phosphate, one of the autophagosome membrane's lipids. Our results reveal how the most upstream ULK1 complex translocates to the autophagosome formation sites during autophagy.

Macroautophagy (hereafter autophagy) is a highly conserved intracellular degradation process induced under stress conditions such as nutrient depletion, infection, hypoxia, and so on[1,2]. Autophagy keeps cellular homeostasis by supplying nutrients and removing harmful materials such as aggregates and damaged organelles. Growing evidence demonstrates that autophagy involves many pathologies, such as cancer, neurodegeneration, infectious diseases, and lifestyle-related diseases[3]. Thus, understanding the molecular mechanisms of autophagy will contribute to developing therapeutic strategies.

Induction of autophagy recruits autophagy-related proteins to a specific subcellular area and generates an isolation membrane/phagophore. The isolation membrane elongates and eventually seals into a double-membraned organelle called autophagosomes, which engulfs cytosolic materials. After fusion with the lysosome, an inner single-membrane containing cytosolic materials is degraded by lysosomal hydrolases[4].

In autophagy initiation, unc51- like autophagy activating kinase 1 (ULK1) complex consisting of ATG13, FIP200/RB1CC1, and ATG101 translocate to the autophagosome formation site from the cytosol. ULK1 complex recruits and activates class III phosphatidylinositol 3-kinase (VPS34) complex, which produces phosphatidylinositol 3-phosphate (PI3P) at the autophagosome formation site. PI3P-binding proteins such as WIPI2 and DFCP1 get recruited. After that, the LC3 protein family is conjugated to phosphatidylethanolamine by ATG16L – ATG5 – ATG12 ubiquitin-like conjugation machinery and located in an elongating isolation membrane. These sequential protein cascades are critical for autophagy induction[5].

[1]Laboratory of Intracellular Membrane Dynamics, Graduate School of Frontier Biosciences, Osaka University, Osaka, Japan. [2]Department of Genetics, Graduate School of Medicine, Osaka University, Osaka, Japan. [3]Integrated Frontier Research for Medical Science Division, Institute for Open and Transdisciplinary Research Initiatives (OTRI), Osaka University, Osaka, Japan. [4]Advanced Light Microscopy Facility, EMBL, Heidelberg, Germany. [5]Cell Biology and Biophysics Unit, EMBL, Heidelberg, Germany. ✉e-mail: tamyoshi@fbs.osaka-u.ac.jp; hamasaki@fbs.osaka-u.ac.jp

ULK1 is a serine/threonine protein kinase and the mammalian orthologue of the yeast Atg1. ULK1 homologs ULK1 and ULK2 are believed to play a crucial role in the autophagy process. They have a degree of redundancy in autophagy, although there are differences between homologs, such as different binding partners[6,7]. Previous studies showed that the membrane association of ULK1 requires its C-terminal domain in mammalian cells and yeast[8–10]. On the other hand, the interaction of ULK1 via the N-terminal region with the endoplasmic reticulum (ER)-resident tail-anchored VAP proteins seems important for the ULK1 retention[11]. Although the translocation of the ULK1 complex is a critical step for initiating autophagy, it was unclear how ULK1 gets anchored to the autophagosome formation site from the cytosol.

ATG14L, originally identified in our previous study, is known to be phosphorylated at Ser by ULK1 upon starvation conditions or inhibition of mTOR[12,13]. The phosphorylation of ATG14L is essential for VPS34 kinase activity and for producing PI3P for autophagosome formation. How the ULK1 complex recruitment to the autophagosome formation site relates to ATG14L phosphorylation remains unclear.

Here we used a siRNA screening and identified a zinc finger DHHC-type palmitoyltransferase 13 (ZDHHC13, also known as HIP14L[14] or HIP3RP) as an essential gene in autophagy. Palmitoylation is a post-translational lipidation reaction of substrate proteins and results in the anchoring of soluble proteins to subcellular membranes or different membrane compartments. In the palmitoylation reaction, acylation occurs on cysteine residues of substrates with fatty acids[15]. The palmitoylation is catalyzed by the zinc finger DHHC-type containing (ZDHHC) family that comprises 24 distinct proteins in mammals. ZDHHC13 is ubiquitously expressed across tissues and localizes to the ER and Golgi apparatus[16,17]. Mutations or deletion of ZDHHC13 results in embryonic lethality, skin and hair abnormalities, osteoporosis, cancers, behavioral abnormalities, and neurodegenerative diseases such as Huntington's disease[18–24]. As cellular functions, ZDHHC13 is known to interact with substrates such as Huntingtin and Drp1[20,25] involved in mitochondrial function[18] or Golgi-phagy[26]. Although these phenotypes are like those observed in autophagy-deficient mice, the role of ZDHHC13 in autophagy is not known.

In this study, we report the newly identified ZDHHC13 palmitoylates ULK1 and recruit its complex to the autophagosome formation site. ULK1 palmitoylation is required for the phosphorylation of ATG14L and autophagy induction. Our study sheds light on a long-standing question of how the ULK1 complex targets autophagosome formation sites and orchestrates autophagy initiation machinery.

## Results
### ZDHHC13 depletion impairs autophagy to bring most upstream ULK1 complex to the formation sites
Since autophagy requires secretion from the ER[27–29], we focused on factors involved in the secretory pathway and performed siRNA screening to identify factors regulating autophagy. To investigate the effect of the siRNA treatment on autophagy, we took two monitoring assays. In one, we measured autophagic activity using HeLa cells stably expressing tandem fluorescent microtubule-associated protein 1 light chain 3 (LC3) (tfLC3, mRFP-EGFP-LC3)[30]. EGFP fluorescence becomes weak under the acidic conditions of the autolysosome, whereas mRFP fluorescence is not affected by pH. Therefore, the EGFP:mRFP ratio correlated with the autophagic flux. Another readout was the monitoring of the number of ATG5 dot formations. ATG5 is recruited to the autophagosome formation site[31], localizing on an isolation membrane/phagophore throughout the formation[5], and released once autophagosome formation completes. The siRNA screening showed that siRNA treatment against ZDHHC13 inhibited autophagy flux and ATG5 puncta formation (Supplementary Fig. 1 and Supplementary Data 1). To confirm the screening results, we tested two siRNAs against ZDHHC13, which reduced endogenous expression and exogenous ZDHHC13-

mNeonGreen (mNG) expression (Supplementary Fig. 2a, b). Inhibitory effects of siZDHHC13 on autophagy flux were observed in both tfLC3 assay and pulse-chase reporter processing assay using Halo-LC3[32] (Fig. 1a, b). To confirm the effect of the knockdown of ZDHHC13, we also generated ZDHHC13 knockout cells using the CRISP-Cas9 system. In all five knockout cell lines, autophagy fluxes were reduced (Fig. 1, Supplementary Fig. 2c). To test the possibility that ZDHHC13 over-expression might enhance autophagy flux, ZDHHC13 wild type (wt) or a catalytic mutant (D453A, Q454A)[33] was transiently expressed in cells expressing Halo-LC3. However, overexpression did not affect autophagy flux significantly (Supplementary Fig. 2d). We next performed a rescue experiment using siRNA-resistant ZDHHC13 to exclude the off-target effect of siRNA treatment. Reintroduction of siRNA-resistant ZDHHC13 wild type (wt) in knockdown cells increased puncta formation of GFP-LC3, whereas ZDHHC13 catalytic mutant did not rescue the knockdown effect (Fig. 1c, Supplementary Fig. 2e). These results show that ZDHHC13 activity has a critical role in starvation-induced autophagy.

We also tested whether ZDHHC13 regulates selective autophagy as well as starvation-induced autophagy. Among selective autophagy, mitophagy and lysophagy were investigated as performed previously[32,34]. In the mitophagy assay, the amount of processed Halo was decreased in ZDHHC13 knockdown cells after 6 h treatment with oligomycin and antimycin (OA) (Supplementary Fig 3a). In the lysophagy assay, GFP-galectin-3 (Gal3) was monitored as a damaged lysosome marker after Leu-Leu methylester hydrobromide (LLOMe) treatment. At 10 h after LLOMe washout, the number of GFP-Gal3 puncta in siZDHHC13 cells was significantly abundant compared to siControl cells (Supplementary Fig 3b), suggesting that ZDHHC13 depletion inhibited the clearance of damaged lysosomes. These results support the idea that ZDHHC13 is required for broad types of autophagy.

We next asked which autophagosome formation process ZDHHC13 regulates. To address this question, we analyzed the effect of ZDHHC13 on known ATG proteins. ZDHHC13 knockdown strongly inhibited the puncta formation of LC3, ATG5, ULK1, and WIPI1 (Fig. 1d). In autophagy induction, ULK1 plays a role in earlier steps than LC3, ATG5, or WIPI1 in the mammalian target of rapamycin (mTOR) signaling-dependent manner. Next, we checked the knockdown effect on mTOR, a protein kinase that negatively regulates autophagy as an upstream key regulator of ULK1. The activity of mTOR was monitored by examining the phosphorylation levels of downstream substrates such as p70S6K, ULK1, and TFEB. In siControl cells treated in a starvation medium showed reduced phosphorylation levels of p70S6K and ULK1 as expected (Fig. 1e). Like in siControl cells, ZDHHC13 knockdown cells also showed a reduction of their phosphorylation levels upon starvation treatment. There is a slight decrease of ULK1 phosphorylation under the nutrient-rich condition but not at a significant level (Fig. 1e). TFEB appears at higher or lower molecular weight based on its phosphorylation status, and no significant differences were observed between ZDHHC13 knockdown cells and control cells. ZDHHC13 knockouts also did not severely affect expression levels of ULK1, FIP200, and ATG13 (Supplementary Fig. 2c). These results suggest that ZDHHC13 mainly regulates autophagy at the post-mTOR step but at an early step as the most upstream ULK1 complex was affected.

### ULK1 is palmitoylated upon autophagy induction
ZDHHC13 is a palmitoyltransferase that catalyzes the addition of palmitate onto substrate proteins[35]. In the palmitoyl-proteomes study using human cells, ULK1 was a candidate for palmitoylated ATG proteins[36]. We first found that ZDHHC13 was bound to ULK1, and such interaction slightly increased under starvation conditions (Fig. 2a). To address the possibility of ULK1-palmitoylation, we next performed a click reaction-based palmitoylation assay. Cells transiently expressing

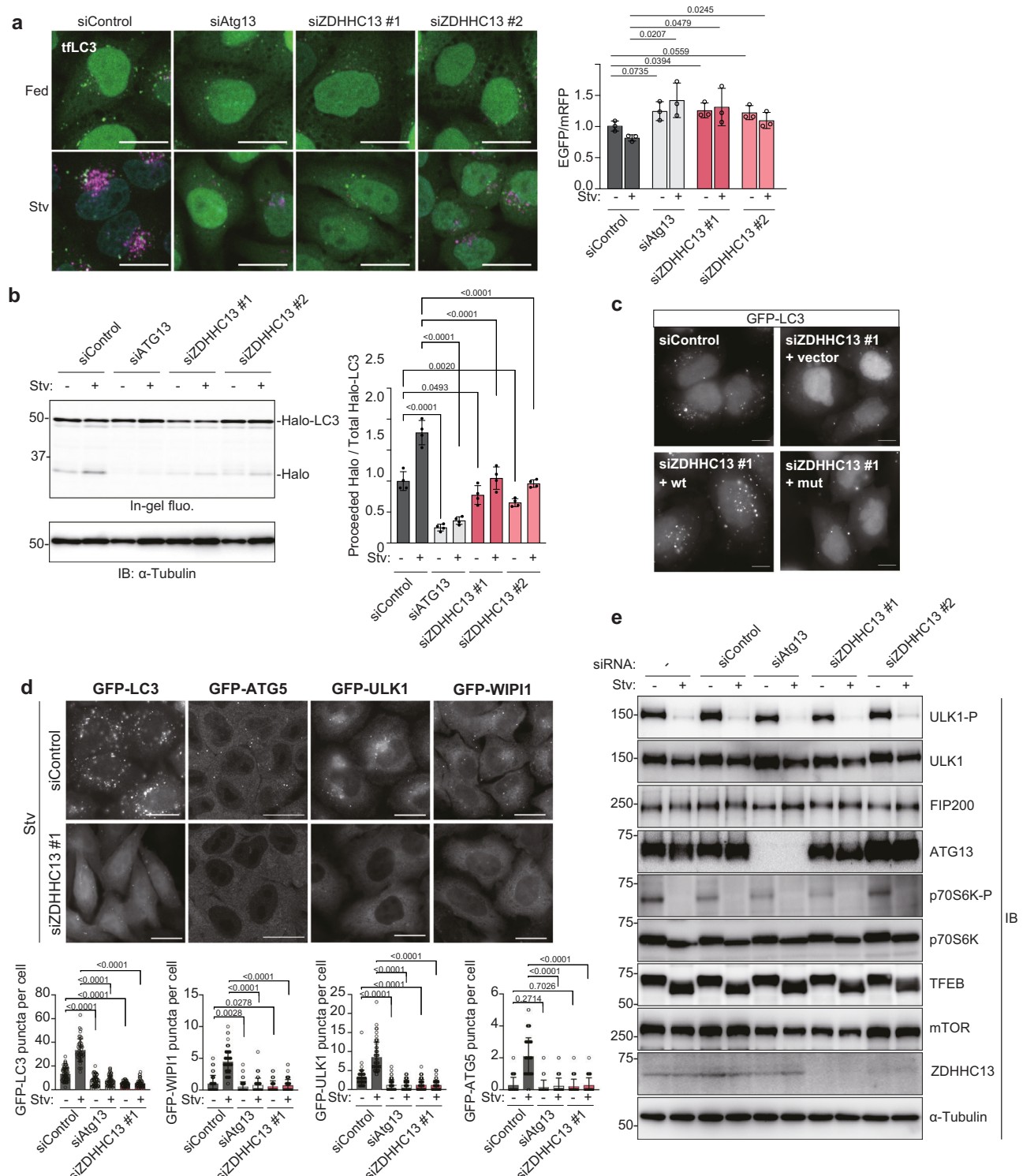

FLAG-tagged ULK1 were incubated with 17-octadecynoic acid (17-ODYA), a commercially available alkynyl fatty acid analog. 17-ODYA is incorporated to substrate proteins and covalently linked to biotin through click chemistry. Biotin-labeled proteins were co-precipitated with streptavidin-magnetic beads and analyzed by immunoblotting. Calnexin (CANX), a palmitoylated protein, was included as a positive control in this assay and showed that it was highly palmitoylated with or without starvation induction. FLAG-ULK1 was also palmitoylated, and such amount increased upon starvation (Fig. 2b and Supplementary Fig. 4a–d). Palmitoylation adds fatty acid, palmitic acid, to the cysteine residue of substrate proteins by thioester bond. We also

confirmed that ULK1 is *S*-acylated through thioester bonds, which are cleaved by hydroxylamine (HAM, NH$_2$OH) treatment (Supplementary Fig. 4b). Palmitoylated ULK1 was not observed in ZDHHC13 knock-down cells (Fig. 2c), suggesting that ZDHHC13 regulates the ULK1 palmitoylation. We also tried to clarify the palmitoylation status of ULK2, ATG13, and FIP200; however, clear precipitated bands for ULK2 and FIP200 were not detected (Supplementary Fig. 4c–e). The palmi-toylation of ATG13 was not conclusive due to non-specific binding to beads both in the click reaction-based palmitoylation assay and in a protein S-palmitoylation detection kit (RapidsPALM, BioDynaics Laboratory Inc.) (Supplementary Fig. 4c, d). The thioester bond of the

**Fig. 1 | ZDHHC13 knockdown impaired autophagy at an early step. a** Monitoring autophagy flux in HeLa cells stably expressing tfLC3. The siRNA-treated cells were incubated in a growth medium (Stv: -) or starvation medium (EBSS, Stv: +) for 4 h. After fixation, EGFP/mRFP intensity was calculated from three independent experiments and shown as mean ± SD. Scale bars indicate 20 µm. A two-tailed unpaired t-test calculated significance. **b** In-gel fluorescence and Immunoblotting of total cell lysates from each knockdown cell stably expressing Halo-LC3. Cells were pulse-labeled for 20 min with TMR-conjugated ligands and incubated in a growth medium or starvation medium for 6 h. Representative images from immunoblotting are shown on the left. The graph is represented as mean ± SD from four experiments. Significance was calculated by one-way ANOVA. **c** HeLa cells stably expressing GFP-LC3 were transfected with siControl or siZDHHC13 and further transfected with a plasmid harboring siRNA-resistant ZDHHC13 wt or catalytic mutant (mut) the next day. After 2 days, the cells were incubated in EBSS for 4 h and subjected to confocal microscopy and immunoblotting. The experiment was performed twice, and representative images are shown. Scale bars indicate 10 µm. **d** ZDHHC13 knockdown inhibited puncta formation of autophagy-related proteins.

HeLa cells stably expressing GFP-LC3, ATG5, ULK1, or WIPI1 were treated with siRNA for 2 days and incubated in EBSS for 4 h. Representative cell images were shown. Scale bars indicate 20 µm. The experiment was performed twice, and representative images are shown. For GFP-LC3 puncta per cells, siControl_Stv-, $n = 66$; siControl_Stv+, $n = 47$; siAtg13_Stv-, $n = 63$; siAtg13_Stv+, $n = 62$; siZDHHC13_Stv-, $n = 64$; siZDHHC13, Stv+, $n = 67$. For GFP-WIPI1 puncta per cells, siControl_Stv-, $n = 71$; siControl_Stv+, $n = 83$; siAtg13_Stv-, $n = 96$; siAtg13_Stv+, $n = 56$; siZDHHC13_Stv-, $n = 65$; siZDHHC13, Stv +, $n = 71$. For GFP-ULK1 puncta per cells, siControl_Stv-, $n = 58$; siControl_Stv+, $n = 51$; siAtg13_Stv-, $n = 55$; siAtg13_Stv+, $n = 58$; siZDHHC13_Stv-, $n = 56$; siZDHHC13, Stv+, $n = 61$. For GFP-ATG5 puncta per cells, siControl_Stv-, $n = 73$; siControl_Stv+, $n = 75$; siAtg13_Stv-, $n = 53$; siAtg13_Stv+, $n = 52$; siZDHHC13_Stv-, $n = 52$; siZDHHC13, Stv+, $n = 59$. A two-tailed paired t-test calculated significance. The exact p values are shown in the figure. **e** ZDHHC13 knockdown did not affect mTOR activity. The cell lysate from siRNA-treated cells was analyzed by immunoblotting. The experiment was performed twice, and representative images are shown.

S-palmitoyl group is cleaved using a high-performance hydroxylamine derivative (hpHA). We confirmed the importance of *S*-acylation through thioester bonds by comparing the hpHA-/MfTag+ sample and hpHA+/MfTag+ (Supplementary Fig. 4d).

### Cys$^{927}$ and Cys$^{1003}$ are the sites of ULK1 palmitoylation

Cysteine residues of substrates are the target for palmitoylation[15]. Previous computational analysis suggested C426, C927, C1003, and C1049 as putative palmitoylation sites of ULK1[37]. In addition, the C-terminal region of ULK1 (*Homo sapiens* aa. 822-1050) has been shown essential for membrane anchoring[8–10]. We therefore tested whether five cysteine residues in the C-terminal region (C927, C950, C1003, C1033, and C1036) are involved in the palmitoylation, excluding C1049 because it is not conserved among mammalian ULK1 proteins and is not essential for membrane anchoring[8]. Our results demonstrated reduced palmitoylation levels in cells expressing ULK1 C927A or C1003A mutants (Fig. 2d), suggesting that C927 and C1003 are likely palmitoylation sites. Notably, yeast Atg1 proteins harbor cysteine residues at the corresponding positions, whereas human ULK2 lacks them, as illustrated by multiple sequence alignment (Fig. 2e). Moreover, in terms of the 3-dimensional structure, all these residues of ULK1 and Atg1 are positioned at the distal ends of the C-terminal helix bundles[38,39], implying that the membrane anchoring takes place on the particular sides of the proteins.

Importantly, C927 and C1003 are not part of the ULK1 C-terminal binding interface with ATG13 and FIP200, as recently revealed by cryo-EM[38] (Fig. 2e), indicating that palmitoylation at these residues does not affect the assembly of the ULK1 complex. Indeed, our immunoprecipitation experiments confirmed that the ULK1 C927A and C1003A mutants could equally pull down both FIP200 and ATG13; thus, the ULK1-FIP200-ATG13 complex formation was unaffected (Fig. 2f–h). Additionally, given that ULK1 is under extensive phosphoregulation[40], the C927A and C1003A mutations did not alter ULK1 phosphorylation status, as all palmitoylation deficient ULK1 CA mutants displayed the same molecular weight in immunoblotting (Fig. 2i). These data support the essential role of ULK1 palmitoylation in anchoring the entire ULK1 complex to the membrane without impacting the complex assembly or ULK1 phosphorylation status.

A previous study reported that ER contact proteins VAPA/B interact with ULK1 and recruit ULK1 complexes on the ER membranes[11]. One question was whether ULK1-VAPs interaction is essential for ULK1 palmitoylation. To address this, we checked the palmitoylation status of the ULK1 Y94A mutant, which is known as a VAPs-binding deficient mutant[11]. The palmitoylation level of ULK1 Y94A was not affected, somewhat enhanced compared to the wild-type (Supplementary Fig. 4f). This suggests that ULK1-VAPs interaction is not essential for ULK1-palmitoylation. The palmitoylation may occur before interaction

with VAPs. Another question is whether the ULK1-VAPs interaction requires ULK1 palmitoylation. ULK1 palmitoylation deficient (C927A, C1003A) mutants still interacted with VAPA and VAPB (Supplementary Fig. 4g), suggesting that ULK1 palmitoylation is independent of the interaction with VAPs.

### ULK1-palmitoylation induces the puncta formation of the ULK1 complex

ULK1/2 is known to be recruited to autophagosome formation sites and localized as dot-like structures. Several previous studies show that ULK1 and ULK2 are functionally redundant[6]. To characterize and monitor only ULK1, we re-expressed ULK1 in ULK1/2 double knockout cells. In HeLa cells stably expressing mNeonGreen (mNG)-ULK1 wild-type, mNG-ULK1 shows dot structures upon starvation. On the other hand, ULK1-palmitoylation deficient CA mutants formed fewer dots under the same condition (Fig. 3a). FIP200 puncta formation was severely affected in cells expressing ULK1 palmitoylation deficient CA mutants (Fig. 3b). There were still a few puncta formed in cells expressing CA mutants; however, the reduction of puncta formation was significant. These remaining FIP200 puncta co-localized with ULK1. These results were consistent with the immunoprecipitation analysis shown in Fig. 2f–h. To further confirm the palmitoylation of ULK1 during autophagy, we treated cells with 2-bromopalmitate (2-BP), a general inhibitor of protein S-palmitoylation. As expected, 2-BP treatment significantly inhibited the puncta formation of ULK1 components and ATG14L in starved cells (Fig. 3c, d). This result supports that ULK1 is palmitoylated during autophagy, and ULK1-palmitoylation is essential for puncta formation.

### Palmitoylation sites in yeast are conserved and essential for the recruitment of Atg1 to pre-autophagosome structure (PAS)

As shown in Fig. 2e, the cysteine residues critical for ULK1 palmitoylation are also conserved in yeast Atg1. To investigate whether the conserved cysteine residues are essential for yeast autophagy, yeast Atg1 wild-type or CA mutants (C731A, C817A) were expressed in the *ΔAtg1* strain. YFP-Atg1 was observed after the nitrogen starvation for 12 h. Wild-type Atg1 showed a dot-like structure, known as PAS, yet Atg1 CA mutants rarely showed dots (Fig. 4). This data suggests the post-translational modification of ULK1/Atg1 during autophagy is conserved from yeast to human.

### Palmitoylation of ULK1 is important for activating the kinase activity against ATG14L

We next asked how the palmitoylation of ULK1 is related to the known autophagy initiation mechanism. ULK1 kinase activity is essential for autophagy and is known to phosphorylate multiple substrates. Although the ULK1 complex, including ULK1, ATG13, and FIP200, are

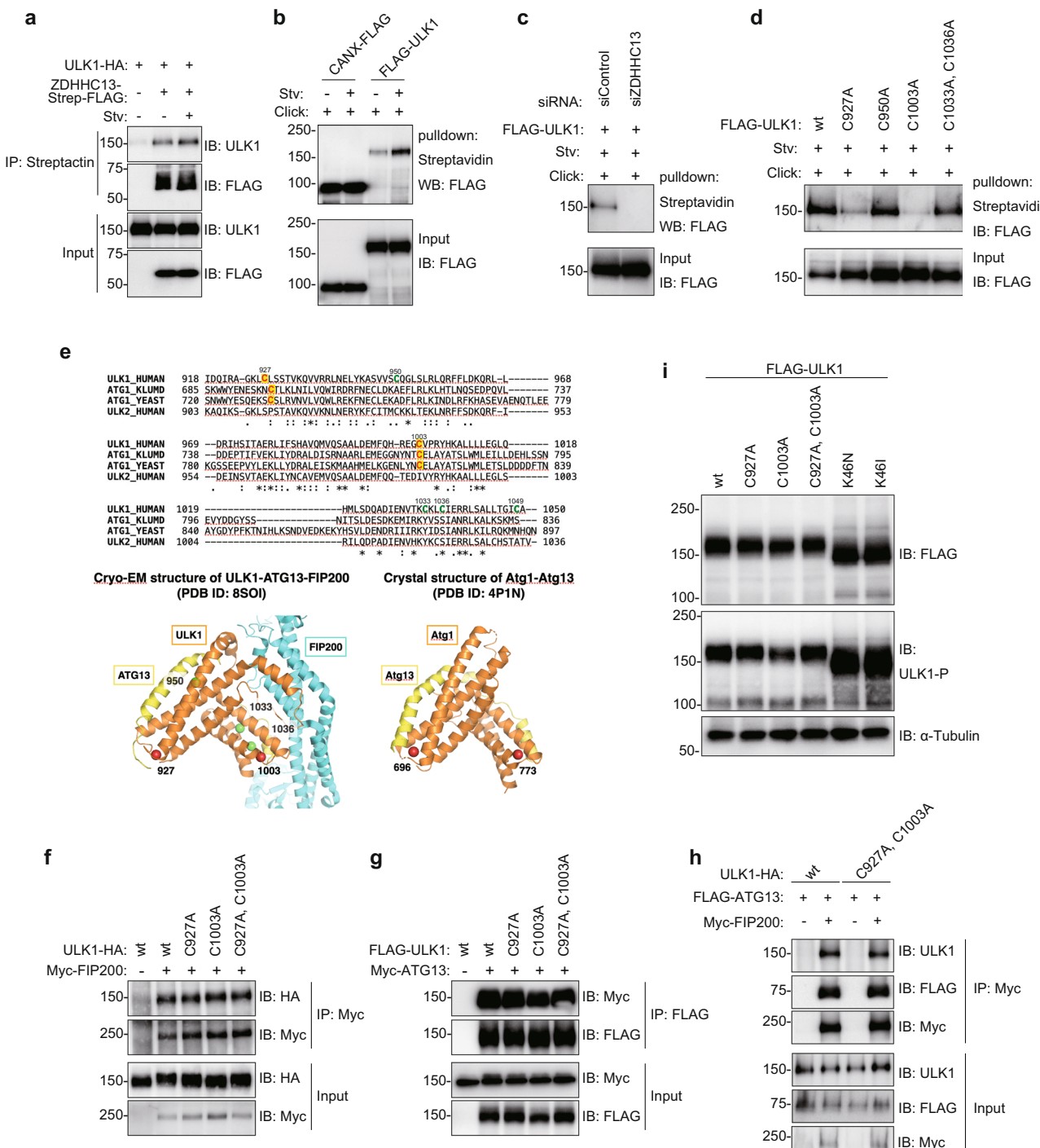

**Fig. 2 | ZDHHC13 regulates ULK1 palmitoylation status. a** Interaction between ZDHHC13 and ULK1. HeLa cells transfected with indicated plasmids were incubated in a growth medium (Stv-) or EBSS (Stv+) and subjected to immunoprecipitation. **b** ULK1 was detected in the palmitoylation assay. HeLa cells transfected with indicated plasmids were incubated in a growth medium or EBSS. The cell lysates were treated as described in Methods. CANX-FLAG was used as a control known to be palmitoylated. Input and pulldown samples were loaded at the following ratios during SDS-PAGE; 1:60 (CANX) and 1:20 (ULK1). **c** ZDHHC13 is required for ULK1 palmitoylation. The siRNA-treated cells were transfected with FLAG-ULK1 plasmid and subjected to palmitoylation assay. **d** C927, and C1003 are involved in ULK1 palmitoylation. As performed in a, HeLa cells transfected with FLAG-ULK1 wild type (wt) or mutant were subjected to palmitoylation assay. **e** (Top) Multiple sequence alignment of C-terminal regions from human ULK1 (O75385), *Saccharomyces cerevisiae* (YEAST) Atg1 (P53104), *Kluyveromyces marxianus* (KLUMD) Atg1 (W0T9X4), and human ULK2 (Q8IYT8). Highlighted in red are C927 and C1003 of ULK1 and

their corresponding cysteine residues of the yeast orthologues, while colored in green are the other cysteines in the ULK1 C-terminal region. (Bottom) Structural comparison between the cryo-EM structure of the human ULK1 complex core (PDB ID: 8SOI, left) and the crystal structure of the *K. marxianus* Atg1-Atg13 complex (PDB ID: 4P1N, right). The cysteines are labeled with the residue numbers and marked with colored spheres as in the multiple sequence alignment. **f–h** ULK1 CA mutations did not affect ULK1 complex formation. **f, g** HeLa cells transfected with indicated plasmids were used for immunoprecipitation as described in Method. **h** ULK1 wt or CA mutant, ATG13 and FIP200 were transiently expressed in ULK1/2 double knockout cells. Cell lysates were analyzed by immunoprecipitation and immunoblotting. **i** ULK1 CA mutants were detected at the same molecular weight as wt. Cell lysates from HeLa cells transiently transfected with each plasmid were analyzed by immunoblotting. For immunoblotting in Fig. 2, the experiment was performed more than twice and representative images are shown.

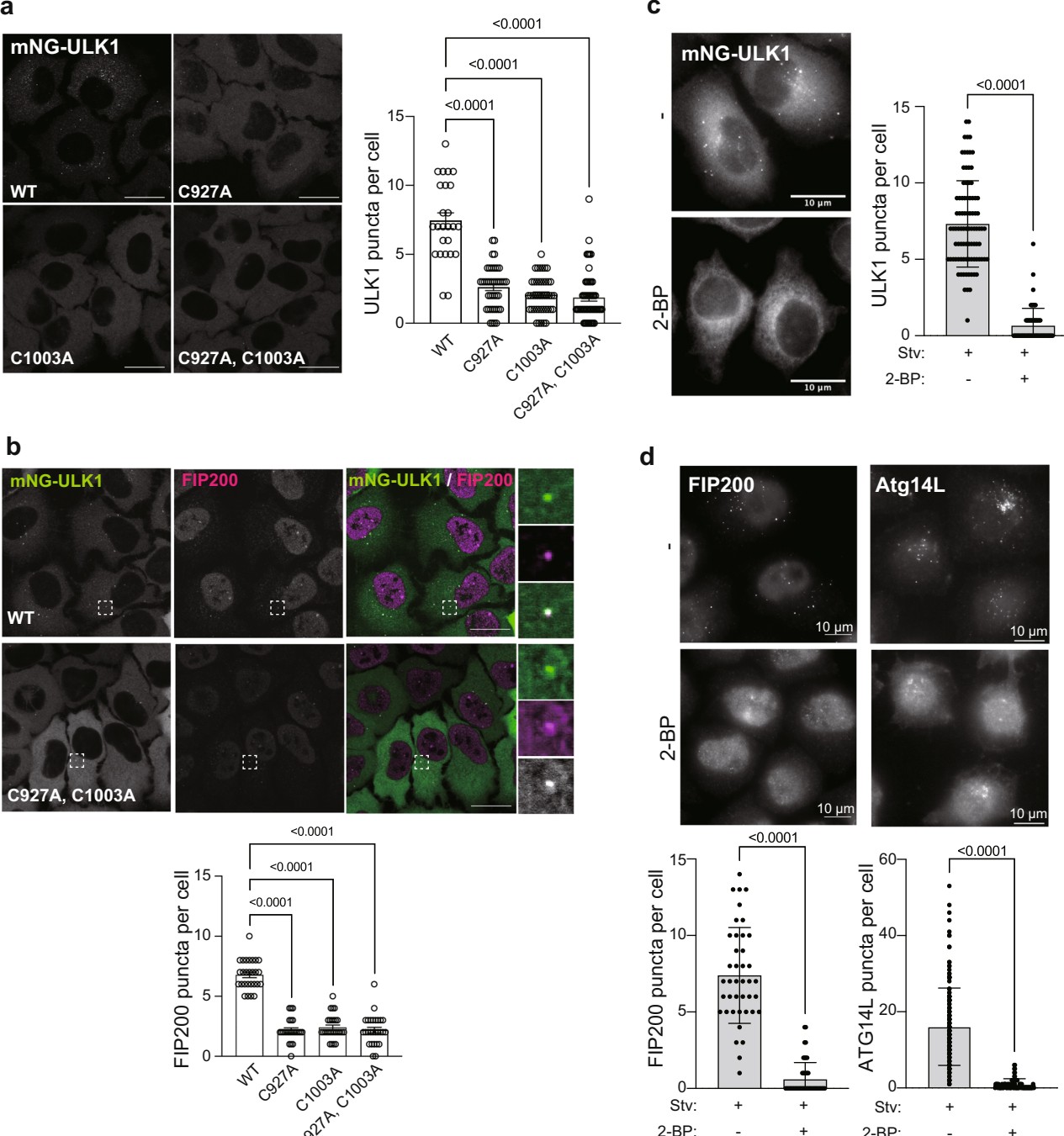

**Fig. 3 | ULK1 puncta formation depends on the palmitoylation. a** ULK1 puncta formation in starvation conditions was decreased in CA mutants. mNeonGreen (mNG)-ULK1 wild-type or each CA mutant was stably expressed in ULK1/2 double knockout HeLa cells. The cells were incubated in EBSS for 4 h, and ULK1 puncta per cell were analyzed. The graph is represented as mean ± SD. Significance was calculated by one-way ANOVA. The experiment was independently repeated twice and showed similar results. WT, n = 25; C927A, n = 43; C1003A, n = 44; C927A, C1003A, n = 54. Representative images are shown. Scale bars indicate 20 μm. **b** Palmitoylation of ULK1 is essential for FIP200 puncta formation. mNG-ULK1 wild-type or each CA mutant was stably expressed in ULK1/2 double knockout HeLa cells. After 4 h of incubation in EBSS, cells were fixed and stained with anti-FIP200 antibodies. Scale bars indicate 20 μm. FIP200 puncta per cell were analyzed. The graph is represented as mean ± SD. Significance was calculated by one-way ANOVA. The experiment was independently repeated twice and showed similar results. WT,

n = 27; C927A, n = 24; C1003A, n = 27; C927A, C1003A, n = 27. **c** ULK1 puncta formation depends on the palmitoylation. ULK1/2 double knockout HeLa cells stably expressing mNG-ULK1 wild-type were treated with DMSO or 200 μM 2-bromoplamitate (2-BP) in EBSS for 3 h. mNG-ULK1 puncta per cell were analyzed and shown as mean ± SD. Significance was calculated by a two-tailed unpaired t-test. The experiment was independently repeated twice and showed similar results. −2-BP, n = 88; +2-BP, N = 55. **d** Puncta formation of endogenous ULK1 complex was impaired by 2-BP treatment. HeLa cells (Left) or HeLa cells stably expressing mCherry-ATG14L (Right) were treated with DMSO or 200 μM 2-BP in an EBSS for 3 h, and stained with anti-FIP200 antibody (Left). The graph shows mean ± SD. A two-tailed unpaired t-test calculated significance. The experiment was independently repeated twice and showed similar results. For number of FIP200 puncta per cell, −2-BP, n = 41; +2-BP, N = 47. For number of ATG14L puncta per cell, −2-BP, n = 165; +2-BP, N = 73.

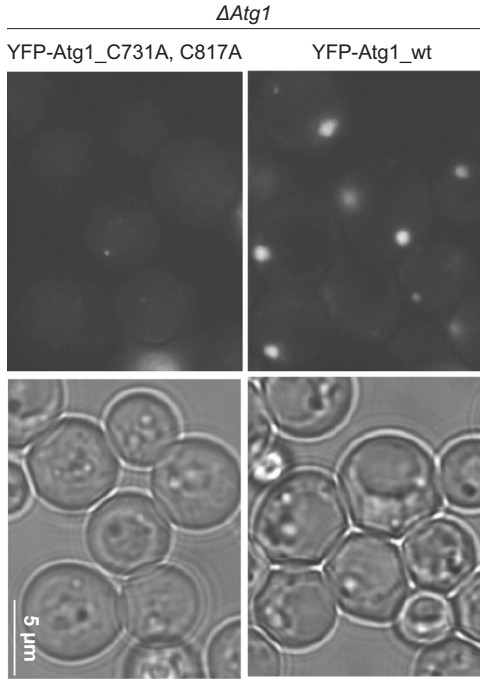

**Fig. 4 | Palmitoylation sites in yeast are conserved and essential for PAS formation.** Conserved cysteine residues in yeast are essential for Atg1 puncta formation. *atg1*Δ cells (BY4741) expressing YFP-Atg1[WT] or YFP-Atg1[C731A, C817A] from single-copy plasmids were treated with rapamycin and analyzed by fluorescence microscopy and differential interference contrast (DIC). The scale bar represents 5 μm. The experiment was independently repeated twice and showed similar results.

reported as substrates, bands of them from immunoblotting were not different in ZDHHC13 knockdown and knockout cells (Fig. 1e and supplementary Fig. 2c). The other downstream substrate is PI3-Kinase complex composed of VPS34, Vps15, Beclin1, and ATG14L. Beclin1 at Ser14 and ATG14L at Ser29 are phosphorylated by ULK1, leading to enhancing VPS34 activity, PI3P production, and autophagy initiation[13,41]. However, the relation between the membrane association of ULK1 and the phosphorylation of ATG14L remained unclear. We tested the effect of the ULK1 palmitoylation deficient CA mutants on the phosphorylation level of ATG14L. Whereas phosphorylation at Ser29 of ATG14L was not detected in ULK1/2 double knockout cells or the double knockout cells expressing kinase-dead mutant (K46I), expression of ULK1 wild-type enhanced the phosphorylation level, which further increased upon starvation treatment (Fig. 5a). These clearly show that phosphorylation of ATG14L at Ser29 depends on ULK1 and its activity. In double knockout cells expressing ULK1 palmitoylation deficient CA mutants, the phosphorylation of ATG14L was significantly lower than wild-type and never enhanced upon starvation treatment. We observed no impact on the phosphorylation status of ATG13, FIP200, or even ULK1, consistent with the result in Fig. 1e. These results suggest that the ULK1-palmitoylation is affecting the kinase activity against ATG14L. A similar experiment was performed to see the effect of the ULK1 CA mutant on the phosphorylation of Beclin1 at Ser14. However, the Beclin1 phosphorylation was not detected with the specific antibody. Also, the treatment with 2-BP in the growth medium reduced the phosphorylation of ATG14L (Fig. 5b). Interestingly, treating with 2-BP in the EBSS medium completely inhibited the phosphorylation of ATG14L.

### Palmitoylation of ULK1 promotes autophagy
As a downstream event of ULK1, phosphorylated PI3-Kinase complex becomes activated and generates PI3P at autophagosome formation sites. To investigate the effect of ULK1 palmitoylation deficiency on the activation of the VPS34 complex, we stained WIPI2, a PI3P-binding protein recruited to autophagosome formation sites during autophagy. The number of WIPI2 puncta was significantly reduced in cells stably expressing ULK1 palmitoylation deficient mutants compared to ULK1 wild-type (Fig. 6a). We also monitored autophagy flux by counting LC3 puncta with or without BafilomycinA1 treatment, which inhibits the lysosomal degradation of LC3. Expression of ULK1 wild-type itself enhanced the number of LC3 dots compared to ULK1/2 double knockout, although ULK1/2 double knockout did not fully impair the LC3 lipidation/dot formation as reported[42–44]. In cells expressing ULK1 palmitoylation deficient CA mutants, the LC3 dot formation was significantly reduced (Fig. 6b). We further confirmed the effect on autophagy flux by pulse-chase reporter processing assay using Halo-LC3 (Fig. 6c). These results suggest that the ULK1 palmitoylation is required for the ULK1 to function and to induce autophagy.

### ZDHHC13 is recruited to the autophagosome formation site during autophagy
ZDHHC13 has 6 transmembrane domains and localizes on intracellular membranes. To define the intracellular distribution under starvation conditions, ZDHHC13-mNG was stably expressed in HeLa cells. Under starvation conditions, 36% of the ZDHHC13 signal was detected as a puncta, although mainly co-localized with Golgi markers (Fig. 7a). 22–44% of these dots were co-localized with stably expressing ULK1, ATG9A, or ATG5 (Fig. 7b). We previously reported that ATG5 is recruited to autophagosome formation sites; ER-mitochondria contact sites[31] suggest that ZDHHC13 is recruited to the autophagosome formation site during autophagy. We also found that ZDHHC13 and ATG9A, a membrane protein mainly localizing on the Golgi and vesicles, were moving together under starvation conditions as vesicles (See supplementary Movie 1). To address the possibility that ZDHHC13 and ATG9A are on the same vesicles, we purified ATG9A-associated membrane fraction from cells stably expressing ATG9A-3xHA. Interestingly, endogenous ZDHHC13 was co-precipitated with ATG9A-associated membranes (Fig. 7c). Furthermore, starvation stress significantly increased the number of ATG9A vesicles positive for ZDHHC13 (Fig.7d). 12.7% or 62.0% of forming autophagosomes labeled with Atg5 and ULK1 were colocalized with ZDHHC13 or ATG9A, respectively (Fig. 7e). These data suggest that the Golgi localized ZDHHC13 is transported to autophagosome formation sites upon starvation induction by Atg9 vesicles (Fig. 7f).

## Discussion
Our results demonstrate that ZDHHC13 palmitoylates ULK1 during autophagy induction, and the ULK1 palmitoylation enhances downstream events such as Atg14L phosphorylation and autophagy. This finding adds new insight into the current model at the early step of autophagy. The phosphorylation of ATG14L by ULK1 contributes to PI3P production by activating VPS34 lipid kinase[13,41], and ATG13 can bind to PI3P through its N-terminus[45]. The ATG13-PI3P binding could contribute to stabilizing the ULK1 complex to the formation site and then onto the phagophore[45,46]. Thus, the palmitoylation of ULK1 functions at an initial step in autophagosome formation.

Regarding the trafficking of ZDHHC13 from the Golgi to the autophagosome formation site, our results support the idea that ZDHHC13 is transported with ATG9A on the same vesicle. We need to find out where ZDHHC13 palmitoylates ULK1; on the ATG9A vesicle or a specific subdomain on the ER membranes in the future. It was not possible to identify with the current imaging technique. Whichever it is, the palmitoylated ULK1 accumulates at the autophagosome formation site when autophagy is induced. This accumulation of palmitoylated ULK1 may create a local environment for activating the downstream events, such as the phosphorylation of ATG14L.

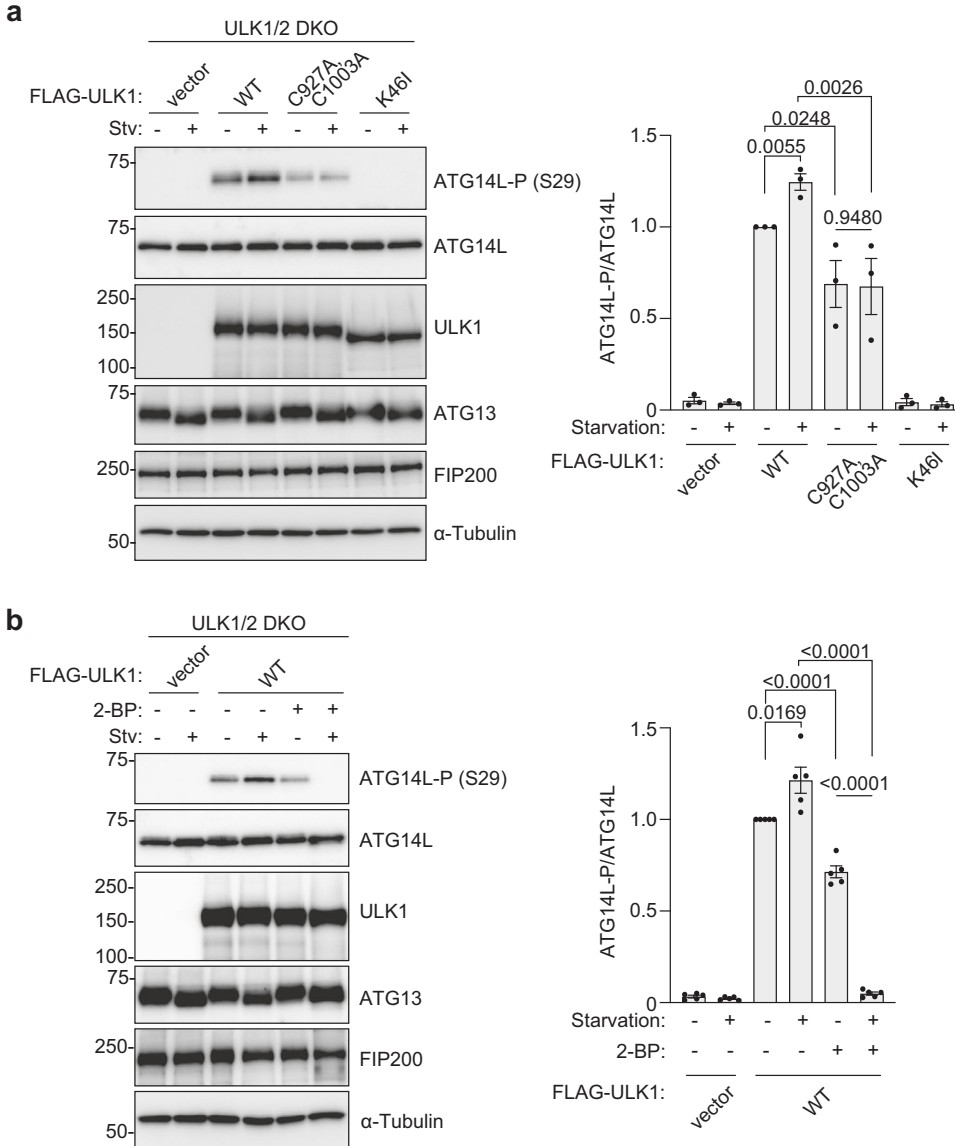

**Fig. 5 | Palmitoylation of ULK1 promotes the kinase activity against ATG14L.**
**a** ULK1/2 double knockout HeLa cells were transiently transfected with empty vector, FLAG-ULK1 wt, FLAG-ULK1 (C927A, C1003A) or FLAG-ULK1 K46I plasmid. After 48 h, cell lysates were used for immunoblotting. Representative images are shown on the left. The graph represents mean ± SD from three experiments. One-way ANOVA and two-tailed unpaired t-test were used to test the significant

difference. **b** ULK1/2 double knockout HeLa cells were transiently transfected with empty vector or FLAG-ULK1 wt plasmid. After 48 h, the cells were treated with 200 μM 2-BP for 3 h. Representative images from immunoblotting are shown on the left. The graph represents mean ± SD from five experiments. One-way ANOVA and two-tailed unpaired t-test were used to test the significant difference.

Monitoring methods should be established in further studies to specify the region of the ULK1 palmitoylation.

Our findings hypothesize that ULK1 and ULK2 are recruited to autophagosome formation sites differently during autophagy, although they have redundant functions to some extent. We found two cysteine residues that are responsible for the ULK1 palmitoylation. These two cysteine residues are conserved in yeast Atg1 but not in ULK2. This fact may support that ULK2 was not precipitated in the click reaction-based palmitoylation assay. The discrepancy in the conservation of these residues suggests that ULK1 and ULK2 are recruited to autophagosome formation sites by different mechanisms.

There are remaining questions to consider; although we demonstrate that ZDHHC13 palmitoylates ULK1 during autophagy induction, the upstream events of the ULK1-palmitoylation are yet to be understood. One speculation would be that the phosphorylation status of ULK1 may be critical for the interaction with ZDHHC13 or for the enzymatic activity. ULK1 is phosphorylated at Ser638 (mouse ULK1

Ser637) and Ser758 (mouse ULK1 Ser757) by mTOR and dephosphorylated under starvation conditions[47]. AMPK and other kinases can also phosphorylate ULK1 at multiple serine residues[6,48,49]. Even ULK1 can be a substrate and auto-phosphorylated at Thr180 by ULK1 itself[50]. Therefore, the timing and these phosphorylation events may be critical in determining the ULK1-palmitoylation. For another hypothesis in upstream events, the enzymatic activity or the binding with substrates of ZDHHC13 can be altered by the post-translational modification such as phosphorylation[51].

Regarding the downstream events of the ULK1-palmitoylation, we found that the phosphorylation status of ATG14L was modified. In further studies, multiple downstream substrates of ULK1 should be focused again on context with the ULK1-palmitoylation. The ULK1-palmitoylation might also function on other downstream substrates.

Atg1 protein is the yeast homolog of mammalian ULK1. The Atg1 comprises five autophagy proteins (Atg1-Atg13-Atg17-Atg29-Atg31) and initiates autophagy[5]. In genetic hierarchical analysis, the

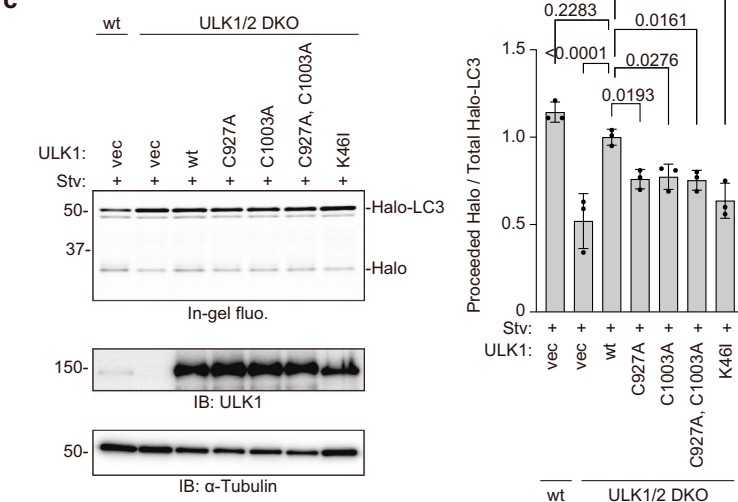

recruitment of the Atg1 complex to the phagophore assembly site (PAS) is independent of other Atg proteins or PI3P production in mammalian and yeast cells[5]. Atg1 consists of 897 amino acids containing a protein kinase domain at the N-terminus, an Atg8-interaction motif (AIM) at the middle, and an early autophagy targeting/tethering (EAT) domain at the C-terminus. The C-terminus of ULK1 is associated with membranes[8]. Consistent with ULK1, the C-terminal EAT domain of

Atg1, which corresponds to the ULK1 C-terminus, binds to liposomes with a preference for small, highly curved vesicles[9,10]. This study shows that potential palmitoylation sites of Atg1, which are conserved residues with ULK1, are located in the EAT domain of Atg1. Our finding supports the idea that the EAT domain of Atg1 binds to membranes directly. However, we cannot exclude the possibility of the involvement of other mechanisms for membrane association. The membrane

**Fig. 6 | The palmitoylation of ULK1 is important for autophagy. a** Palmitoylation of ULK1 enhances the class III PI3K activation. mNG-ULK1 wild-type or each CA mutant was stably expressed in ULK1/2 double knockout HeLa cells. After 4 h of incubation in EBSS, cells were fixed with MeOH and stained with anti-WIPI2 antibody as described in Methods. Alexa 568-labeled WIPI2 signals were shown as green. Scale bars indicate 20 μm. WIPI2 puncta per cell was analyzed by Fiji. The graph is represented as mean ± SD. Significance was calculated by one-way ANOVA. The experiment was independently repeated twice and showed similar results. WT, -, $n = 28$; DKO, -, $n = 47$; DKO, WT, $n = 41$; DKO, C927A, $n = 42$; DKO, C1003A, $n = 35$; DKO, C927A, C1003A, $n = 39$. **b** Palmitoylation of ULK1 is essential for efficient LC3 puncta formation. mNG-ULK1 wild-type or each CA mutant was stably expressed in ULK1/2 double knockout HeLa cells. After 4 h of incubation in EBSS with or without BafA1, cells were fixed and stained with anti-LC3 antibody as described. Alexa 568-labeled LC3 signals were shown as green. Scale bars indicate 20 μm. LC3 puncta per cell was analyzed by Fiji. The graph is represented as mean ± SD. Significance was calculated by one-way ANOVA. The experiment was independently repeated twice and showed similar results. WT, -, -BafA1, $n = 27$; WT, -, +BafA1, $n = 16$; DKO, -, -BafA1, $n = 24$; DKO, -, +BafA1, $n = 16$; DKO, WT, -BafA1, $n = 17$; DKO, WT, +BafA1, $n = 14$; DKO, C927A, -BafA1, $n = 25$; DKO, C927A, +BafA1, $n = 13$; DKO, C1003A, -BafA1, $n = 26$; DKO, C1003A, +BafA1, $n = 19$; DKO, C927A, C1003A, -BafA1, $n = 25$; DKO, C927A, C1003A, +BafA1, $n = 21$. **c** Palmitoylation of ULK1 is important for efficient autophagy flux. HeLa wt or ULK1/2 double knockout cells stably expressing Halo-LC3 were transfected with empty vector, ULK1 wt, or mutant plasmid. The cells were analyzed by the pulse-chase reporter processing assay as described in Methods. ULK1 expression levels and loading were analyzed in immunoblotting. Representative images from immunoblotting are shown on the left. The graph is represented as mean ± SD from three experiments. Significance was calculated by one-way ANOVA.

association of Atg1 is also regulated through the AIM of Atg1[52,53]. ATG13/Atg13 has a putative lipid binding ability, although the domain containing responsible arginine/lysine residues is not conserved[45]. Seven DHHC cysteine-rich domain-containing proteins (DHHC proteins) are identified as yeast acyltransferase for palmitoylation reaction. Vac8 is known to be palmitoylated by Pfa3[54] and tethers the Atg1 complex to the vacuolar membrane[55]. Further studies are needed to find responsible proteins and molecular mechanisms in yeast autophagy.

## Methods

### Reagents and resources
All reagents, resources, and antibodies used in this study are listed in Supplementary Table 1. Bafilomycin A1 (BafA1), 17-ODYA (Alk-16), Azide-PEG3-biotin conjugate, TBTA, and 2-BP, LLOMe, oligomycin, and antimycin were dissolved in DMSO for stock solutions. TCEP and $CuSO_4$ were dissolved in water.

### Plasmids
All plasmids used in the study were listed in Supplementary Table 1. Full-length sequences of human ZDHHC13, human VAPA, and human VAPB were amplified with gene-specific primers listed in Table S1 from HeLa cDNA and cloned into pMRX-ires-puro_EGFP or pMRX-ires-puro_mNG vector. A site-direct mutagenesis was used to generate ULK1 mutants and ZDHHC13 catalytic mutants.

### Cell culture and transfection
All cell lines used in this study are listed in Supplementary Table 1. Cells were maintained in Dulbecco's modified Eagle medium (DMEM), supplemented with 2 mM L-glutamine, nonessential amino acids, 100 U/ml penicillin, 100 μg/ml streptomycin, and 10% fetal bovine serum. Transduced cells were selected in a medium containing appropriate antibiotics, as shown in Supplementary Table 1. To induce autophagy and monitor autophagy flux, cells were incubated with serum and amino acid deprived medium (EBSS) with or without 200 nM BafA1 for 4 h at 37 °C. For DNA transfection, TransIT-LT1 Transfection Reagent, or polyethylenimine (PEI-MAX), was used according to the manufacturer's protocol. Lipofectamine RNAiMAX transfection regent was used according to the manufacturer's protocol for siRNA transfection.

### Retrovirus and lentivirus production
Retrovirus production was performed as described earlier[56]. In brief, Plat-E cells were co-transfected with the envelope-encoding plasmid pLP-VSVG and a pMRX vector plasmid containing the gene-of-interest using polyethylenimine (PEI MAX). Supernatants were harvested at 48 h post-transfection.

Lentivirus production and cell transductions were performed exactly as described earlier[57]. In brief, Lenti-X 293 T cells were co-transfected with the packaging plasmid pCMV-dR8.91, the envelope-encoding plasmid pMD2.G, and a CRISPR-Cas9 plasmid. Supernatants were harvested 48 and 72 h post-transfection and filtered.

### Generation of knockout cell lines
The short guide RNA (sgRNA) targeting gene of interest are listed in the Supplementary Data 1. To generate knockout cell lines, cells were transduced with a given lentivirus, and 2 days later, cells were cultured in medium containing 3 μg/ml puromycin for at least 3 days. Knock-out was validated by immunoblotting and used in experiments.

### Generation of cell lines stably expressing exogenous proteins
The supernatants containing retrovirus were filtered and added to a cell culture medium with 4 μg/ml polybrene. After inoculation overnight, the medium was replaced with fresh medium. After 2 days, cells were cultured in a medium containing 3 μg/ml puromycin or blasticidin for at least 3 days. The expression of the exogenous protein was validated by immunoblotting or fluorescent microscopy.

### siRNA screening in HeLa cells stably expressing tfLC3
The siRNA transfection solution was prepared as reported previously[58]. Briefly, into each well of a 96-V-shaped plate, 5 μl of siRNA stock solutions, with a concentration of 3 μM, were added to 7 μl of transfection reagent solution (3 μl of OptiMEM containing 0.4 M sucrose + 1.75 μl of $H_2O$ + 1.75 μl Lipofectamine 2000) and mixed thoroughly. After 20 min of incubation at RT, 7 μl of 0.2% gelatin solution were added to each well, and the resulting transfection mixes were diluted 1:50 with water (1 + 49 μl, respectively). The siRNA transfection solution (50 μl) was distributed into empty glass-bottom 96-wells imaging plates (Greiner, Item-No. 655891)[58] and lyophilized. HeLa cells stably expressing tfLC3 or GFP-ATG5 were seeded into the 96 well imaging plate at 2,000 cells per well using Multidrop Combi (Thermo Scientific) cell seeding device. The cells were incubated for 60 h at 37 °C. Afterwards the cells were incubated in a growth medium or EBSS for additional 4 h at 37 °C. For fixation, the cells were treated with 4% paraformaldehyde in PBS containing Hoechst 33342 nuclear staining dye. All images were automatically acquired with an Olympus Scan^R screening microscope using a UPlanApo NA 0.7 dry objective[58].

### Pulse-chase reporter processing assay and in-gel fluorescence imaging
The pulse-chase reporter processing assay was performed as shown previously[32]. To monitor starvation-induced autophagy flux, HeLa cells stably expressing Halo-LC3 were incubated with 100 nM tetramethylrhodamine (TMR)-conjugated ligands for 20 min at 37 °C. After washing out, the cells were incubated in a starvation medium (EBSS) for 6 h at 37 °C. The cells were lysed in 2x sample buffer and subjected to SDS-PAGE. The gel was visualized with ChemiDoc imaging system (BioRad). A mitophagy assay was also performed, as shown previously[32]. To investigate parking-mediated mitophagy,

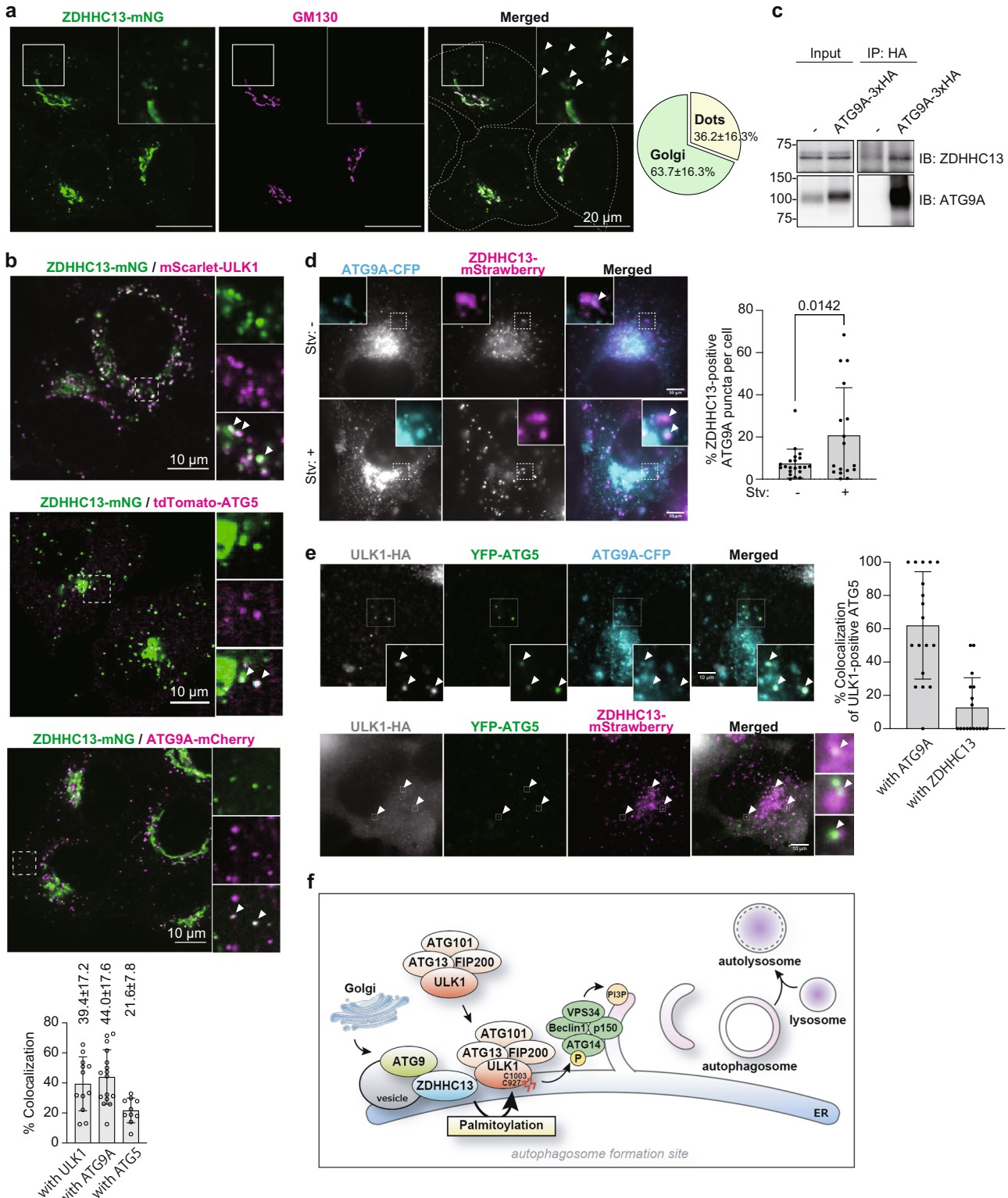

myc-Parkin, and pSu9-Halo-mGFP were stably expressed in HeLa cells. The cells were incubated with TMR-conjugated ligands for 20 min at 37 °C, and then washed out. The cells were treated with 1 μM oligomycin and 2 μM antimycin (OA) for 6 h.

**Palmitoylation assay**

$2.0 \times 10^6$ HeLa cells were seeded into a 10 cm dish and cultured overnight. Cells were transfected with pcDNA3.1-3xFLAG-ULK1 or pcDNA3.1-Calnexin (CANX)- FLAG using PEI-MAX. After 24 h, the

medium was replaced with a fresh growth medium containing 25 μM 17-ODYA. At 48 h post-transfection, the cells were incubated in a growth medium or starvation medium (Stv, EBSS), which was supplemented with 50 μM 17-ODYA for 2 h to stimulate autophagy. The cells were lysed with PBST buffer (1% Triton-X100, 1x protease inhibitor cocktail, and 1 mM PMSF in PBS). Supernatants were mixed with click reaction solution at final concentrations of 1 mM $CuSO_4/5H_2O$, 1 mM TCEP, 100 μM TBTA, and 500 μM biotin-azide incubated for 1 h at room temperature. For hydroxylamine (HAM) treatment, final 2.5%

**Fig. 7 | ZDHHC13 is recruited to the autophagosome formation site.**
**a, b** Colocalizations of ZDHHC13 and autophagy-related proteins during starvation-induced autophagy. **a** Localization of ZDHHC13-mNG. Cells stably expressing ZDHHC13-mNG were stained with anti-GM130, a Golgi marker. Areas with white rectangles are magnified in each image. Scale bars indicate 20 μm. The experiment was repeated twice, and the percentage of colocalization was calculated from a total of 38 cells and shown as a pie chart. **b** HeLa cells stably expressing ZDHHC13-mNG and autophagy-related protein were incubated in EBSS for 4 h and fixed. The cells were observed by a confocal microscope and shown as representative images. White arrowheads indicate representative colocalizations. Scale bars indicate 10 μm. The graph shows as mean ± SD. ULK1, $n = 12$; ATG9A, $n = 16$; ATG5, $n = 10$. **c** Endogenous ZDHHC13 was co-precipitated with ATG9A-associated membranes. Cells stably expressing ATG9A-3xHA were incubated in EBSS for 4 h and subjected to immunoprecipitation with anti-HA magnetic beads. Parental HeLa cells were used as a control sample. The experiment was performed twice, and representative images are shown. **d, e** Colocalizations of ZDHHC13 with autophagy-related proteins were temporally affected upon starvation induction. ULK1-HA was transiently expressed in COS7 cells, stably expressing YFP-ATG5, ATG9-CFP, and ZDHHC13-mStrawberry. After 24 h, the cells were incubated in a growth medium or EBSS for 2 h or 4 h and fixed. The fixed cells were stained with anti-HA antibody. The graph shows mean ± SD. A two-tailed unpaired t-test calculated significance. Stv-, $n = 21$; Stv, $n = 17$. White arrowheads indicate representative colocalizations. **e** Percentage of ULK1-positive ATG5 puncta overlapping with ATG9A was 62.0 ± 31.4% (e, upper panels). The percentage of ULK1-positive ATG5 puncta overlapping with ZDHHC13 was 12.7 ± 17.4% (**e**, lower panels). ATG9, $n = 18$; ZDHHC13, $n = 17$. Scale bars: 10 μm. **f** Schematical model summarizing the role of ZDHHC13 in autophagy. Upon autophagy induction, ZDHHC13 is recruited to autophagosome formation sites together with ATG9A. ULK1 is palmitoylated at Cys[927] and Cys[1003] residues by ZDHHC13, and the ULK1 complex are anchored to an autophagosome formation site. The palmitoylation of ULK1 promotes phosphorylation of ATG14L, which leads to VPS34 activation. Activated PI3-kinase complex produces PI3P at autophagosome formation sites. These sequential reactions trigger efficient autophagy induction. Parts of this figure were produced using images from Servier Medical Art. Servier Medical Art is licensed under a CC BY 4.0 license https://creativecommons.org/licenses/by/4.0/.

---

HAM was added in the click reaction solution and incubated for 1 h at room temperature. After centrifugation at $20k \times g$, 4 °C for 5 min, supernatants were transferred to new tubes and incubated with 20 μl of magnetic Streptavidin-agarose beads for 1 h at 4 °C. The beads were washed with PBST buffer four times using a magnetic separation rack. For protein elution, the beads were mix with 30 μl of 2x sample buffer (100 mM Tris-HCl [pH 6.8], 4% SDS, 12% β-mercaptoethanol, 20% glycerol, 0.001% bromophenol blue). After 60 min incubation, supernatants were transferred to a new tube. These samples were subjected to immunoblotting without sample boiling.

Palmitoylation assay with protein S-palmitoylation detection kit (RapidsPALM, BioDynamics Laboratory Inc.) performed as described in manufacturer's protocol. HeLa cells were transiently transfected with FLAG-ULK1 plasmid and incubated in starvation medium (EBSS) for 4 h at 48 h post-transfection. Each sample from one 10 cm dish was split to two groups and analyzed; cleavage⁻, MfTag-labeling (hpHA/MfTag: -/+) and cleavage⁺, MfTag-labeling (hpHA/MfTag: +/+).

### Immunofluorescence microscopy
Immunofluorescence microscopy was performed as described previously[57]. Cells cultured on glass coverslips were fixed with 4% paraformaldehyde in PBS for 30 min. The cells were permeabilized with PBS containing 0.1% Triton X-100 or 50 μg/ml digitonin, blocked with 5% FBS or 0.2% gelatin, and then incubated with diluted primary antibody for 60–120 min at room temperature. After washing with PBS three times, the cells were incubated with Alexa- or ATTO-dye labeled secondary antibodies in PBS containing 5% FBS or 0.2% gelatin for 60 min. The coverslips were mounted with a mounting medium (VECTASHIELD), and images were obtained with the FV1000 confocal microscope (OLYMPUS) or IX83 widefield microscope (OLYMPUS). WIPI2 staining was performed as described previously[59]. Cells were fixed with ice-cold methanol for 10 min and washed with PBS three times. The cells were incubated in permeabilization buffer (0.1% saponin, 0.5% BSA, and 50 mM $NH_4Cl$ in PBS) for 60 min. An anti-WIPI2 antibody is diluted with blocking buffer (0.1% saponin and 0.5% BSA in PBS) and incubated for 60 min at room temperature. After washing with PBS three times, the cells were incubated with secondary antibody diluted with blocking buffer for 60 min.

### Immunoprecipitation and immunoblotting
Immunoprecipitation was performed as described previously[57]. Cells were lysed with lysis buffer (50 mM Tris [pH 7.5], 150 mM NaCl, 1% TritonX-100 and 1x protease inhibitor cocktail). After centrifugation at $20k \times g$, 4 °C for 10 min, supernatants were transferred to new tubes and incubated with 30 μl of anti-FLAG-M2 agarose beads, anti-HA magnetic beads, or anti-FLAG magnetic beads for 2 h at 4 °C. The beads were washed with lysis buffer four times, mixed with 30 μl of 2x sample buffer (100 mM Tris-HCl [pH 6.8], 4% SDS, 12% β-mercaptoethanol, 20% glycerol, 0.001% bromophenol blue) and incubated at 95 °C for 5 min. Supernatants were transferred to new tubes and subjected to immunoblotting. For the interaction between ZDHHC13 and ULK1, cells were lysed with lysis buffer (50 mM Tris [pH 7.5], 150 mM NaCl, 2% DDM and 1x protease inhibitor cocktail). Supernatants after centrifugation were incubated with 20 μl of streptactin beads, for 1 h at 4 °C. The beads were washed with wash buffer (50 mM Tris [pH 7.5], 500 mM NaCl, 2% DDM) four times, mixed with 30 μl of 2x sample buffer. Protein samples were not heated at 95 °C to avoid aggregation of ZDHHC13 protein.

For purification of ATG9A-associated membranes, we modified a Lyso-IP protocol described previously[60]. Briefly, we were wash HeLa cells stably expressing ATG9A-3xHA twice and collected the cells by centrifugation at $2000 \times g$ for 2 min after treatment with for 4 h. The cells were resuspended in KPBS buffer (136 mM KCl, 10 mM $KH_2PO_4$) and homogenized with 27G syringe by 30-50 strokes. After the centrifugation of the cell lysate at $2000 \times g$ for 10 min, the supernatant was mixed with anti-HA magnetic beads for 5 min. The beads were washed 5 times with KPBS, and proteins associated with ATG9A-3xHA were eluted with 2x sample buffer.

Proteins were separated by SDS-polyacrylamide gel electrophoresis for immunoblotting and electro-transferred onto PVDF membranes. After blocking the membranes with 5% nonfat milk, they were incubated overnight at 4 °C with diluted primary antibodies. After washing with 0.5% Tween 20 in PBS, membranes were incubated with diluted secondary horseradish peroxidase-conjugated antibodies for 1 h at room temperature. Membranes were developed using Immobilon Forte Western HRP substrate (Merk), and signals were detected by ChemiDoc Touch (Bio-Rad).

### Autophagy induction and observation of Atg1 puncta in yeast
BY4741 ΔAtg1 cells were transformed with pR316-YFP-Atg1(wt) and C731A/C817A mutants. Both strains were cultured to O.D.600 = 1, then cultured in SD(-N) medium for 12 h. Cells were placed on the cover slip and observed by using Olympus IX83.

### Multiple sequence alignment and structural comparison
The amino-acid sequences of human ULK1 (O75385), *Saccharomyces cerevisiae* Atg1 (P53104), *Kluyveromyces marxianus* Atg1 (W0T9X4), and human ULK2 (Q8IYT8) were retrieved and aligned online at the UniProtKB website (https://www.uniprot.org/). The cryo-EM structure of the human ULK1 complex core (PDB accession code: 8SOI) and the

crystal structure of the *K. marxianus* Atg1-Atg13 complex (PDB accession code: 4P1N) were aligned and visualized using the PyMOL Molecular Graphics System, Version 2.5.4 (Schrödinger, LLC).

## Statistics and reproducibility

Unless otherwise stated, values represent the mean of a given number of replicates. Error bars are SD as indicated in the figure legends. Student t-tests and one-way ANOVA were performed using Prism9 software (GraphPad software), and a $P < 0.05$ was considered statistically significant. $P$ values from statistical analysis were shown in each graphs. All experiments were repeated more than twice independently, as indicated in the figure legends. Representative images are shown from immunoblotting, in-gel fluorescence or microscopy. No statistical method was used to predetermine sample size. The experiments were not randomized, and the investigators were not blinded to allocation during the experiments.

## Reporting summary

Further information on research design is available in the Nature Portfolio Reporting Summary linked to this article.

## Data availability

Source data are provided with this paper.

## Code availability

No new algorithms were developed for this manuscript. All code generated for analysis is available from the authors upon request.

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

## Acknowledgements

We thank Dr. Yamaoka for the gifts of pMRX-ires-puro vector and pMRX-ires bsr vector; Dr. Kitamura (University of Tokyo, Japan) for the gift of Plat-E cells. We thank Dr. Kihara for the gift of calnexin cDNA. We also thank Dr. Mizushima for gifts of pMRX-IP-HaloTag7-LC3 and pMRX-IB-pSu9-Halo-mGFP. We acknowledge the excellent technical assistance of Akiko Nezu and Giovanni Bravin. K.T. is supported by JST CREST, Ono Medical Research Foundation, UCL-OU seed fund, JSPS KAKENHI 21K06169, and MSD Life Science Foundation. T.K. is supported by JST CREST and JSPS KAKENHI 20K05839. T.Y. is supported by Grant-in-Aid for Scientific Research on Innovative Areas JP2511001 and JSPS KAKENHI JP26251020 22H04982 M.H. is supported by Grant-in-Aid for Scientific Research on Innovative Areas JP20H05239 and JSPS KAKENHI JP15H04371 and JP21K06152.

## Author contributions

K.T., K.I., M.H., and T.Y. designed the study. K.I., F.V., R.P., B.N., C.T., and S.R. conducted siRNA screening. K.T., K.I., H.K., T.F., K.F., and K.Y. performed experiments using cultured cells such as fluorescent microscopy, immunoblotting, and immunoprecipitation. M.H. performed yeast experiments. T.K. analyzed amino acid sequence and protein structure. K.T., K.I., T.K., and M.H. wrote the manuscript.

## Competing interests

The authors declare no competing interests.
