## [Peer Review File · Nature Communications]

Palmitoylation of ULK1 by ZDHHC13 plays a crucial role in autophagyEditorial Note: This manuscript has been previously reviewed at another journal that is not operating a transparent peer review scheme. This document only contains reviewer comments and rebuttal letters for versions considered at *Nature Communications*.

REVIEWER COMMENTS

Reviewer #2 (Remarks to the Author):

The Authors have assessed in this submission all possible concerns that were previously raised, and the paper is now suitable for publication as it stands.

Reviewer #3 (Remarks to the Author):

The authors have addressed all my concerns and I find the manuscript is acceptable for publication.

Reviewer #4 (Remarks to the Author):

In the revised manuscript entitled "Palmitoylation of ULK1 by ZDHHC13 plays a crucial role in autophagy", the authors have skillfully answered at the questions raised by the reviewers, often providing additional data. However, the concerns of reviewers #1 and #3 regarding the temporal recruitment of ULK1 on autophagosome and its palmitoylation and the precise effect/relevance of palmitoylation on the complex formation and autophagy activation remain. The mechanism of ZDHHC13DH activation/recruitment after autophagy activation to palmitoylate ULK1 is also uninvestigated. More importantly, despite the reported role of autophagy in multiple systems, the manuscript lacks evidence for a pathophysiological relevance of the mechanism while relying heavily on protein/mutant overexpression.

Data in a biological context using endogenous proteins to explain the upstream regulation of the palmitoylation would strengthen the enthusiasm.

Minor points:

-Both ULK1 and ULK2 phosphorylate ATG14 on Ser 9, which is used as a mechanism for the relevance of ULK1 palmitoylation, the use of ULK1/ULK2 double KO does not help to understand this question.

-The western blot (Fig1e) lacks the membrane showing ZDHHC13 levels after silencing (the targeted protein) which is crucial for the interpretation. The upregulation (Fig1e) of ATG13 by one of the ZDHHC13 siRNAs suggests additional side effects.

-Throughout the manuscript, the authors should provide (eg, in the figure legend?) the number and sample size used for the statistics used; this is most important in a manuscript relying on quantification of images/foci/puncta.

Responses to reviewers

We would appreciate your comments. We have addressed each of the points raised by the reviewers below. The Reviewers' comments are shown in Italics and our responses are written in blue.

Reviewer #2 (Remarks to the Author):

The Authors have assessed in this submission all possible concerns that were previously raised, and the paper is now suitable for publication as it stands.

We thank the reviewer for their time and thoughtful review. We are grateful for their positive feedback and acceptance of the manuscript.

Reviewer #3 (Remarks to the Author):

The authors have addressed all my concerns and I find the manuscript is acceptable for publication.

We appreciate your time and thoughtful review, and we are grateful for their positive feedback and acceptance of the manuscript.

Reviewer #4 (Remarks to the Author):

In the revised manuscript entitled "Palmitoylation of ULK1 by ZDHHC13 plays a crucial role in autophagy", the authors have skillfully answered at the questions raised by the reviewers, often providing additional data. However, the concerns of reviewers #1 and #3 regarding the temporal recruitment of ULK1 on autophagosome and its palmitoylation and the precise effect/relevance of palmitoylation on the complex formation and autophagy activation remain.

Thank you for the comments. To elucidate the mechanism by which ZDHHC13 palmitoylates ULK1 during autophagy initiation, we investigated the interaction of ZDHHC13 with ULK1 following starvation treatment. As shown below, the starvation treatment resulted in a slight increase in their binding. While the enhanced binding suggests it promotes the palmitoylation reaction, we are not

excluding the possibility of additional regulatory mechanisms that may control this reaction during autophagy initiation. We replaced to the new data for Figure 2a.

Figure 2a: Interaction between ZDHHC13 and ULK1. ULK1-HA and ZDHHC13-Strep-FLAG were transiently expressed in HeLa cells. After 48h, the cells were incubated in a growth medium (-Stv) or EBSS (+Stv) for 4h and subjected to immunoprecipitation with streptactin beads.

To address the temporal recruitment of ULK1 to the autophagosome, we analyzed the colocalization of ZDHHC13 with autophagy proteins. First, we found that the colocalization of ATG9A with ZDHHC13 was significantly increased under starvation conditions (Fig. 7d). We also found that 12.7% of forming autophagosomes labeled with Atg5 and ULK1 were colocalized with ZDHHC13 in starved cells (Fig. 7e). These data support the idea that ZDHHC13 is temporally recruited to autophagosome formation sites together with ULK1 and ATG9A upon starvation induction and included in Fig. 7.

Figure 7d and e: Colocalizations of ZDHHC13 with autophagy-related proteins were temporally affected upon starvation induction. ULK1-HA was transiently expressed in COS7 cells, stably expressing YFP-ATG5, ATG9A-CFP, and ZDHHC13-mStrawberry. After 24h, the cells were incubated in a growth medium or EBSS for 2h or 4h and fixed. The fixed cells were stained with anti-HA antibody. Cell images were acquired by OLYMPUS IX83 microscope using a 100x objective lens. The percentages of colocalization from more than 17 cells per experiment were analyzed using Fiji. The graph shows mean \pm SD. The percentage of ULK1-positive ATG5 puncta overlapping with ZDHHC13 was $12.7 \pm 17.4\%$ (e, upper panels). The percentage of ULK1-positive ATG5 puncta overlapping with ATG9A was $62.0 \pm 31.4\%$ (e, lower panels). Significance was calculated by unpaired t-test.

To address the effect of ULK1-palmitoylation on autophagy activation, we investigated the effect of 2-BP treatment on the puncta formation of ATG14L. As shown below, ATG14L puncta significantly decreased under starvation treatment.

This result supports that ULK1-palmitoylation is essential for autophagy initiation. This data is included in Figure 3d.

Figure 3d: Puncta formation of endogenous ULK1 complex was impaired by 2-BP treatment. HeLa cells stably expressing mCherry-ATG14L were treated with DMSO or 200 μ M 2-bromoplmitate (2-BP) in an EBSS medium for 3 h. Cell images were acquired by OLYMPUS IX83 microscope at a 60x objective lens. The graph shows mean \pm SD. Significance was calculated by unpaired t-test. The experiment was independently repeated twice and showed similar results. More than 41 cells from two independent experiments were analyzed.

The mechanism of ZDHHC13DH activation/recruitment after autophagy activation to palmitoylate ULK1 is also uninvestigated.

We acknowledge the reviewer's interest in the activation mechanism of ZDHHC13 under autophagy inhibition conditions. While this specific question was not raised during the initial review process, we recognize its potential significance in elucidating the molecular underpinnings of the observed mechanism. However, given the current scope of this manuscript and the limitations in definitively addressing this point within the existing data, we discussed with the editor and have opted to focus primarily on the core findings and their immediate implications in this study. We agree that understanding the activation mechanism of ZDHHC13 under these conditions represents a compelling avenue for future research.

More importantly, despite the reported role of autophagy in multiple systems, the manuscript lacks evidence for a pathophysiological relevance of the mechanism while relying heavily on protein/mutant overexpression.

We appreciate the reviewer's suggestion regarding exploring the potential pathophysiological role of the ZDHHC13-ULK1 pathway in autophagy. While this aspect was not explicitly raised during the first round of review, we acknowledge its potential interest in furthering the understanding of the mechanism described in this study. However, given the current scope of this manuscript and the limitations in addressing this point definitively within the context of the present work, we discussed with the editor, and we have opted to focus on the core findings and their immediate implications in this study. Suppose there is a pathophysiological role of the ZDHHC13-ULK1 pathway. In that case, it represents an intriguing avenue for future research, and we need to wait for upcoming studies to clear this aspect in greater detail.

Data in a biological context using endogenous proteins to explain the upstream regulation of the palmitoylation would strengthen the enthusiasm.

We appreciate the reviewer's thoughtful comment regarding the use of endogenous proteins. We acknowledge the limitations of commercially available antibodies against ZDHHC13 for immunofluorescent microscopy applications. Our attempts to generate an antibody specific to the palmitoylated form of ULK1 have also failed. Consequently, we could not address the question of upstream regulation using these specific endogenous proteins through immunofluorescent microscopy. Alternative approaches and methodologies are needed to overcome these limitations.

Minor points:

-Both ULK1 and ULK2 phosphorylate ATG14 on Ser 9, which is used as a mechanism for the relevance of ULK1 palmitoylation, the use of ULK1/ULK2 double KO does not help to understand this question.

Since ULK1 and ULK2 are reported to phosphorylate ATG14L (reference-13 in manuscript; Park et al., Autophagy 2016, PMID: 27046250), we will only know the importance of ULK1-Palmitoylation if we put the ULK1-CA mutant back into ULK1,2 DKO. To specifically isolate the effects of ULK1 and investigate the role of its palmitoylation, we employed a ULK1/2 double knockout cell line, re-

introducing either wild-type or CA-mutant of ULK1. We hope this explanation clarifies the reasoning behind it.

-The western blot (Fig1e) lacks the membrane showing ZDHHC13 levels after silencing (the targeted protein) which is crucial for the interpretation. The upregulation (Fig1e) of ATG13 by one of the ZDHHC13 siRNAs suggests additional side effects.

We appreciate the reviewer's suggestion regarding the ATG13 protein level in knockdown samples. In response, we have included an immunoblot image (Fig. 1e) demonstrating ZDHHC13 levels in knockdown samples. Regarding the upregulation of ATG13, we acknowledge that observations across knockdown cells (#1 and #2) and knockout cells (#1-#5) are only partially consistent. As the reviewer mentioned, the increase in siZDHHC13 #2-treated cells may be attributed to additional off-target effects.

Figure 1e: ZDHHC13 knockdown did not affect mTOR activity. Knockdown cells were incubated in a growth medium (Stv: -) or EBSS (Stv: +). The cell lysate was analyzed by immunoblotting with indicated antibodies. The experiment was performed twice, and representative images are shown.

-Throughout the manuscript, the authors should provide (eg, in the figure legend?) the number and sample size used for the statistics used; this is most important in a manuscript relying on quantification of images/foci/puncta.

We are grateful for the reviewer's suggestion regarding additional information. In response, we have included the missing information within the figure legends, such as the number of replicates and sample size used in the statistical analysis.

REVIEWERS' COMMENTS

Reviewer #4 (Remarks to the Author):

The Authors have addressed in this resubmission most of the concerns previously raised by the review process, and the paper is now suitable for publication.